

# Social presence effect in language comprehension: evidence from event-related potential (ERP) research

Teng Yu[*], Xue Sui[*] and Yu tong Li

School of Psychology, Liaoning Normal University, Dalian, Liaoning Province, China
[*] These authors contributed equally to this work.

## ABSTRACT

**Objective**. This study aimed to examine the impact of social presence on Chinese reading comprehension and associated neural responses.

**Methods**. Participants tasked with reading Chinese sentences either alone or in the presence of others and subsequently assessing the accuracy of the sentences' meanings. Concurrently, we recorded the participants' electrical brain responses to critical word processing.

**Results**. Behavioral results indicated no significant effect of social presence on the judgment of sentence accuracy. Electroencephalogram (EEG) results, however, revealed that reading in the presence of others elicited more pronounced left anterior negativity (LAN) components in the left front of the scalp compared to reading alone. Additionally, incorrect meanings triggered larger N400 and P600 amplitudes in the mid-parietal region than correct meanings.

**Conclusion**. Social presence intensifies early neural responses during the reading of Chinese sentences, although it does not influence semantic integration or conflict resolution. These findings support the notion that social context affects language processing.

## INTRODUCTION

Speech serves as a medium for communication, wherein participants must leverage shared world knowledge, which relies on mutual understanding (*Clark & Marshall, 1981*). The aforementioned processes are influenced by egocentric biases, subsequently followed by a delayed adjustment process (*Barr & Keysar, 2006*; *Keysar, 2007*). Adjustments are flexibly made based on the peer's perspective or characteristics to enhance communication (*Brennan, Galati & Kuhlen, 2010*). Therefore, speech communication studies should consider the dynamic interplay and mutual influence among interlocutors (*Brennan, Galati & Kuhlen, 2010*), an aspect often overlooked in research focused solely on individual cognitive processes. At present, the vast majority of speech studies are done by subjects alone, ignoring the social attributes of language use. When other people are present or working with others to complete a task, the efficiency of an individual's behavior changes. It is pertinent to explore whether the mere presence of others, without direct task involvement,

Corresponding author
Yu tong Li, dearliyutong@163.com

influences cognitive functions. While early research indicated that the mere presence of others altered behavior (*Burnham, 1910*), this finding was not replicated in later studies (*Zajonc, 1965*). Therefore, the potential social facilitation effect of mere presence remains under debate.

The event-related potential (ERP) technique, known for its high temporal resolution, is extensively used to study the temporal dynamics of text reading. Key electroencephalogram (EEG) components related to text reading include left anterior negativity (LAN), N400, and P600. The LAN is a negative waveform occurring within a 100–500 ms time window. This component is associated with morphological and syntactic violations (*Yang et al., 2021*). Furthermore, the processing of correct yet uncommon grammatical structures can elicit larger LAN amplitudes (*Hoen & Dominey, 2000*; *Hagoort, 2003*). The N400 component, peaking around 400 ms in the central parietal lobe, reflects the difficulty of the integration of vocabulary and context (*Kutas & Federmeier, 2011*). It is also sensitive to social contexts, evidenced by the so-called ''social N400 effect'' (*Rueschemeyer, Gardner & Stoner, 2015*; *Jouravlev et al., 2019*; *Westley, Kohút & Rueschemeyer, 2017*). P600, manifesting as a positive shift from 500–1,000 ms, primarily indicates late-stage semantic and syntactic integration and reanalysis (*Molinaro, Barber & Carreiras, 2011*; *Burkhardt, 2007*; *Hagoort, 2003*). Additionally, some researchers posit that P600 may reflect conflicts between an individual's perceptions and established communication models (*Kuperberg, Brothers & Wlotko, 2020*).

Some researchers (*Rueschemeyer, Gardner & Stoner, 2015*), investigated the effects of social presence on language comprehension, discovering that even the mere presence of others could influence subjects' language comprehension. Specifically, the perspectives of speakers and listeners reciprocally influenced their understanding of language. When in the company of others, subjects extended their processing beyond their own perspective to include that of their peers. However, the study by *Rueschemeyer, Gardner & Stoner (2015)* only demonstrated that under explicit experimental instructions and the presence of others, subjects more actively considered the viewpoints of others. The impact of a non-participatory presence on speech comprehension remains a topic of keen interest among researchers.

*Hinchcliffe et al. (2020)* were pioneers in addressing this issue. They established two experimental conditions—alone or in the presence of others—and tasked subjects with reading sentences and judging their correctness. These sentences fell into three categories: both semantically and syntactically correct, semantically correct but syntactically incorrect, and syntactically correct but semantically incorrect. Their findings indicated that social presence did not alter behavioral outcomes. However, significant differences emerged in EEG results; subjects reading alone showed greater LAN amplitude in anterior and central regions for syntactically incorrect sentences. In contrast, this effect shifted to central and posterior regions when others were present, indicating a more centrally distributed parietal LAN. The processing of semantically incorrect sentences elicited greater N400 amplitude in the central region, regardless of social presence, but additional N400 activity was noted in the right anterior region when others were present. The presence of others also heightened activity in the precuneus. The researchers attributed the shift in LAN location

to a change in individual strategies for addressing syntactic errors—from algorithmic to more heuristic—and linked increased precuneus activity to social cognition and attention.

There are few studies on the influence of others on speech comprehension, and most of the studies on social facilitation theory are carried out in the background of European and American culture. European and American cultures are mostly individualistic, while Chinese culture is collectivist. Different cultures lead to different behavior patterns and thinking patterns (*LeFebvre & Franke, 2013*; *Mazar & Aggarwal, 2011*). In the context of collectivist culture, whether the presence of others has an impact on speech comprehension remains to be explored. In addition, research on speech comprehension frequently centers on phonetic characters, yet studies on ideographic characters are sparse. Phonetic characters, primarily from the Indo-European family, markedly differ from ideographic characters of the Sino-Tibetan family (*Liu et al., 2023*). Moreover, the dynamics between semantic and syntactic processing vary significantly. Some studies have found that the syntactic processing of German and other phonetic characters mostly follows syntactic priority (*Friederici, 2011*). Syntactic priority theory holds that sentence processing takes syntactic processing as the core and syntactic processing takes precedence and is independent of semantic processing. However, the processing of ideographic characters such as Chinese mostly follows the semantic-centered theory (*Friederici, 2002*; *Trueswell, Tanenhaus & Garnsey, 1994*; *Yu & Zhang, 2008*). Semanticocentrism holds that syntax is of limited help to sentence understanding, and semantic processing dominates the process of sentence understanding. In addition, Chinese, a typical ideogram, lacks morphological variations (*Li, Bates & MacWhinney, 1993*). In contrast, alphabetic writing display pronounced morphological change. Consequently, findings from alphabetic writing studies are not directly applicable to ideographic characters. Based on the above two points, it is necessary to further explore whether the simple social situation of the presence of others can affect the Chinese speech processing and the neural mechanism behind it in the context of collectivist culture.

This study examines the effect of external presence on Chinese reading comprehension at both behavioral and neural levels. Two conditions were established: subjects alone and subjects with others present. Participants were tasked with reading Chinese sentences of either correct or incorrect meaning and subsequently determining their accuracy. The correctness of the sentences hinged on the proper use of keywords placed at the end, with corresponding electrical brain activity recorded. In the presence condition, subjects were informed that others were engaged in a memory task, though sentence evaluations were conducted independently (*Hagoort, 2003*). As such, no behavioral response differences were anticipated. Thus, the study posits that external presence influences Chinese reading comprehension primarily through neural responses. The second hypothesis suggests that early electrical responses in subjects are altered, considering sentences were presented as others completed a memory task. Since the bystanders did not participate in the reading tasks, the third hypothesis posits that EEG disparities during the integration and processing stages of Chinese comprehension are primarily attributable to material differences.

## MATERIALS & METHODS

### Participants

A sensitivity power analysis conducted using G* Power 3.1 (*Faul et al., 2009*) indicated that 24 participants were necessary to detect a medium effect size ($f = 0.25$) with a power of 80% (two-tailed $\alpha = 0.05$, 1 - $\beta = 0.80$) in a mixed-design with repeated measures involving two independent variables. Thirty right-handed, Chinese-speaking undergraduate students with normal or corrected-to-normal vision participated (*Oldfield, 1971*) None had a history of neurological and language disability. Due to excessive eye movements or EEG artifacts, four participants were excluded, resulting in a final sample 26 participants (10 males, mean $22.11 \pm 3.47$ years) for data analysis. All the participants written informed consent prior to the experiment, which received approval from the Ethics Committee of Liaoning Normal University (LL2024114).

### Experimental design and material

This study employed a 2×2 within-subject design, with the first independent variable being the presence of others (alone *vs.* presence of others) and the second being sentence type (semantically correct *vs.* incorrect).

In this study, the fixed syntactic structure of "A sentence containing the word '把'(pinyin: ba)" was used as experimental material, controlling for the influence of syntactic structure. The experiment utilized 200 experimental sentences and 50 filler sentences, presented through E-prime across two blocks containing 50 semantically correct sentences, 50 semantically incorrect sentences, and 25 filler sentences each. Keywords in semantically correct sentences were altered to create incorrect versions. Sentences ranged from 12 to 14 words in length. A 7-point scale assessed keyword familiarity, sentence difficulty, and overall sentence rationality. Paired sample T-tests revealed no significant difference in keyword familiarity ($t$ (15) $= -1.633$, $p > 0.05$) or sentence difficulty ($t$ (15) $= -2.809$, $p > 0.05$) between the two sentence types, but a significant difference in sentence rationality ($t$ (16) $= 195.624$, $p < 0.05$) was noted. For examples of the experimental materials see Table 1.

### Experimental procedure

Participants sat approximately 70 cm from a liquid crystal display (LCD) screen, evaluating the semantic correctness of sentences presented word-by-word under both social conditions. In the solitary condition, the participant performed the task alone. Conversely, in the collaborative condition, a confederate was positioned approximately 55 cm to the right of the participant within the participant's peripheral vision, yet without visible reactions. The experiment featured two confederates one female and one male, both undergraduate students unacquainted with the participants and matching their gender. Confederates acted as assistants, simulating word memorization for a vocabulary recognition test. Upon the confederate's arrival in the collaborative condition, participants were instructed to read sentences and assess their semantic accuracy, while the confederate memorized the words for a subsequent test. To enhance realism, researchers demonstrated a sample memory questionnaire. Prior to the main experiment, participants engaged in six
**Table 1  Examples of experimental materials.**

| Sentence type | Example |
| --- | --- |
| Correct sentences | 李明 把 刚买的 种子 种进**花盆**。 |
| | Li Ming planted the seeds he had just bought into **a flower pot** |
| Incorrect sentences | 李明把刚买的种子种进**板凳**。 |
| | Li Ming planted the seeds he had just bought into **the bench** |

Notes.
These examples were originally in Chinese. Critical words have been bolded.

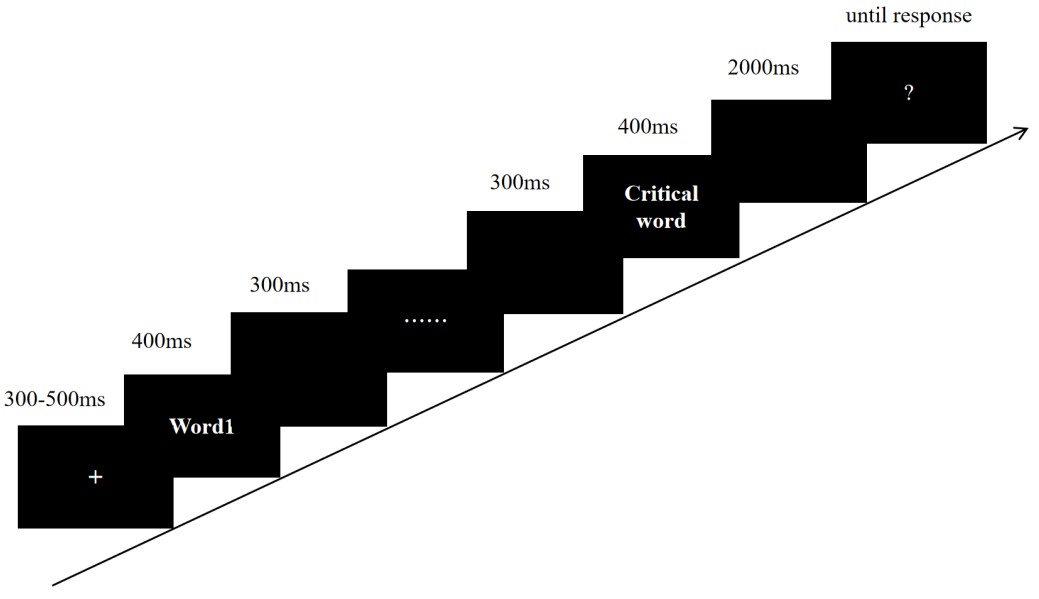

**Figure 1    The example for procedure of a trial.**

practice trials to familiarize themselves with the instructions. They sequentially completed the two social conditions, with their order counterbalanced among participants, and a 10 min interval between them.

As depicted in Fig. 1, each trial initiated with a fixation cross for a 300-500 ms jitter duration, followed by a word-by-word sentence display at the center of the monitor. Each word appeared for 400 ms, separated by a 300 ms inter-stimulus interval. After the sentence, a blank screen appeared for 2,000 ms, succeeded by a question mark until the participant responded by pressing a designated key (F/J for correct/incorrect, counterbalanced across subjects). All text was presented in white on a black background using 24-point Song typeface. To avoid ocular artifacts during EEG recordings, participants were requested not to blink between sentences. The EEG response to noun processing at sentence end was recorded.

## EEG recording and preprocessing

The EEG was captured using a 64-Channel Brain Products system. The FPz electrode served as the ground, and FCz electrode as the reference. Vertical electrooculogram (EOG) was

monitored below the right eye to detect eye movements and blinks. Electrode impedances were maintained below 5 kΩ, and EEG signals were digitized at 1,000 Hz with a 0.1–100 Hz bandpass.

Raw EEG data were processed offline in Brain Vision Analyzer version 2.0 (Brain Products, GmbH; Gilching, Germany), re-referenced to the average reference of left and right mastoids, and filtered between 0.1–30 Hz (24 dB/oct slope). Ocular artifacts were corrected *via* independent component analysis (ICA), utilizing the Informax Restricted ICA algorithm from Analyze2.0 software. EEG data were decomposed into 64 components, which were examined semi-automatically for component relevance, specifically excluding those associated with horizontal eye movements and blinks. On average, 4.10 components (SD = 1.17) were excluded per subject. EEG data were segmented into 1,000 ms epochs centered on the critical word stimulus, baseline-corrected to 200 ms pre-stimulus. Epochs with peak-to-peak deflections exceeding ±100 µV were discarded. After the removal of incorrect trials and artifact-ridden epochs, the average numbers of usable trials was 39.30 (SD = 7.89, range: 16–50) for the semantic correctness in the collaborative condition; 35.35 (SD = 7.82, range: 15–48) for semantic incorrectness in the collaborative condition, 36.31 (SD = 9.70, range: 18–49) for semantic correctness in the solitary condition, and 33.88 (SD = 8.68, range: 15–49) for semantic incorrectness in the solitary condition.

## Statistical analysis

We collected response time and accuracy data for each participants' tasks. Response times associated with incorrect answers or those deviating more than 2.5 standard deviations from the mean were excluded from the analysis. Two factor repeated-measures ANOVAs were conducted on response times and accuracy, examining the effects of social condition (presence, alone) and semantic correctness (correct, incorrect).

Informed by prior research (*Hinchcliffe et al., 2020*; *Rueschemeyer, Gardner & Stoner, 2015*; *Westley, Kohút & Rueschemeyer, 2017*), we selected specific EEG components for analysis: the left-anterior LAN (300–500 ms), the N400 (300–500 ms) and the P600 (600–800 ms) components at both central and parietal sites. Figure 2 illustrates the regions of interest, including the left-anterior (F1, F3, FC3, FC5), middle-anterior (Fz, FCz, FC1, FC2), right-anterior (F2, F4, FC4, FC6), left-central (C3, C5, CP1, CP3), middle-central (C1, Cz, C2 , CPz), right-central (C4, C6, CP2, CP4), left-posterior (P1, P3, PO3), middle-central (Pz, POz), and right-posterior (P2, P4, PO4)sites. We calculated the amplitudes of each component within these regions based on the average amplitudes during specified time windows. We calculated the amplitudes of each component within these regions based on the average amplitudes during specified time windows. A repeated-measures ANOVA was applied for the LAN component at anterior site considering factors such as social condition, semantic correctness, and hemisphere. For N400 and P600 components, we performed two repeated-measure ANOVA for each component involving the within factors social condition, semantic correctness and hemisphere. This two repeated-measure of ANOVA were conducted for central and parietal sites respectively. For all ANOVA with multiple degree of freedom, the Greenhouse-Geisser correction was applied to adjust for violation of sphericity. When necessary, Bonferroni corrections were used to address
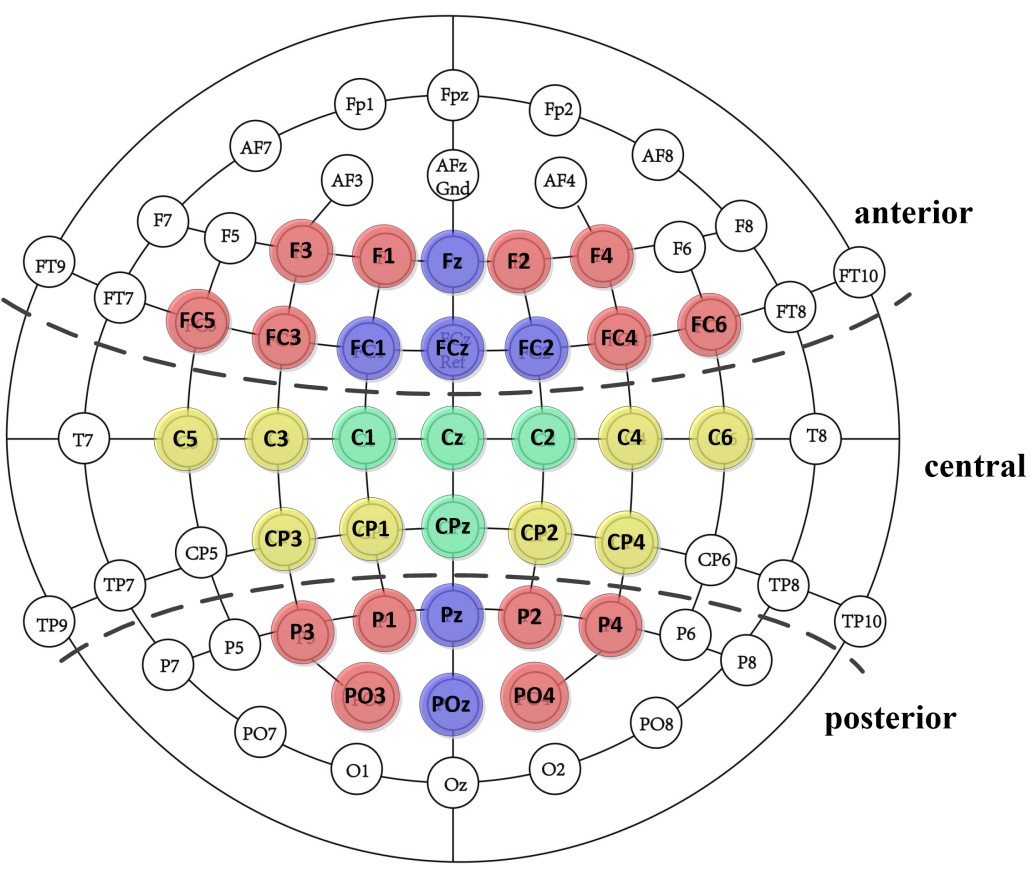

**Figure 2** Electrode layout on the scalp.

multiple comparisons issues. We set the alpha level at 0.05 (two-tailed) for all inferential analyses. Significant interactions were further explored using a simple effects model. We reported only significant main effects of social condition and semantic correctness, and significant interactions involving topographic factors with at least one experimental factor.

# RESULTS

## Behavioral results

Behavioral data for each condition are detailed in Table 2. The rmANOVA on accuracy indicated a significant main effect for semantic correctness, $F(1, 25) = 38.38, p < 0.001, \eta_p^2 = 0.61$, with higher accuracy for semantically incorrect sentences. However, no significant main effect for social condition, $F(1, 25) = 0.95, p = 0.34$, or interaction between social condition and semantic correctness, $F(1, 25) = 0.05, p = 0.83$, was observed.

The rmANOVA on response times revealed no significant main effects for social condition, $F(1, 25) = 0.32, p = 0.58$, or semantic correctness, $F(1, 25) = 1.78, p = 0.19$. Additionally, the interaction between social condition and semantically correctness were not significant, $F(1, 25) = 1.23, p = 0.28$.

**Table 2   The response time and accuracy for the four conditions.**

| measure | Presence condition | | Alone condition | |
|---|---|---|---|---|
| | **Correct** | **Incorrect** | **Correct** | **Incorrect** |
| **Response time (ms)** | $520 \pm 135$ | $516 \pm 121$ | $519 \pm 135$ | $500 \pm 111$ |
| **Accuracy (%)** | $94.46 \pm 2.91$ | $87.84 \pm 3.67$ | $93.24 \pm 3.32$ | $86.08 \pm 5.13$ |

## ERP results

Figure 3 displays the grand average waveforms of ERPs in different sites. For the LAN component, the results of rmANOVA indicated a main effect of semantic correctness, $F(1, 25) = 31.30$, $p < 0.001$, $\eta_p^2 = 0.56$. Correct sentences elicited smaller LAN amplitudes than incorrect sentences. There was also a significant interaction between semantic correctness and hemisphere, $F(2, 50) = 7.00$, $p = 0.002$, $\eta_p^2 = 0.22$. Correct sentences elicited smaller LAN amplitudes in the left ($\Delta = 0.73$, 95% CI [0.34–1.12], $p = 0.001$), middle ($\Delta = 1.22$, 95% CI [0.80–1.65], $p < 0.001$), and right area ($\Delta = 1.02$, 95% CI [0.65–1.40], $p < 0.001$). Additionally, a three-way interaction between semantic correctness, social condition and hemisphere was observed, $F(2, 52) = 3.38$, $p = 0.04$, $\eta_p^2 = 0.12$. The results indicated that presence condition elicited larger LAN amplitudes than the alone condition for incorrect sentences only in left frontal area ($\Delta = -1.07$, 95% CI [$-1.89$–$-0.25$], $p = 0.01$).

For the N400 in the central region, a significant main effect of semantic correctness was noted, $F(1, 25) = 36.45$, $p < 0.001$, $\eta_p^2 = 0.59$, showing that correct sentences elicited smaller N400 amplitudes than incorrect sentences. Furthermore, an interaction between semantic correctness and hemisphere was significant, $F(2, 52) = 18.33$, $p < 0.001$, $\eta_p^2 = 0.42$. Correct sentences elicited smaller N400 in the left ($\Delta = 0.71$, 95% CI [0.32–1.11], $p = 0.001$), middle ($\Delta = 1.52$, 95% CI [1.02–2.01], $p < 0.001$), and right area ($\Delta = 1.25$, 95% CI [0.87–1.62], $p < 0.001$). Similarly, the results for the N400 component in the parietal region demonstrated a significant main effect of semantically correctness, $F(1, 25) = 7.73$, $p = 0.01$, $\eta_p^2 = 0.24$, indicating that correct sentences elicited smaller N400 amplitudes. An interaction between semantic correctness and hemisphere were also significant, $F(2, 52) = 5.75$, $p = 0.01$, $\eta_p^2 = 0.19$. Correct sentences elicited smaller N400 amplitudes in the middle ($\Delta = 0.94.52$, 95% CI [0.36–1.52], $p = 0.003$) and right areas ($\Delta = 1.04$, 95% CI [0.34–1.74], $p = 0.005$).

For the P600 in the central region, only a significant interaction between semantic correctness and hemisphere was found, $F(2, 52) = 7.68$, $p = 0.001$, $\eta_p^2 = 0.24$, with no significant between correct sentences and incorrect sentences in any area (all $ps > 0.08$). The results of the P600 in the parietal region showed a significant main effect of semantic correctness, $F(1, 25) = 12.41$, $p = 0.002$, $\eta_p^2 = 0.33$, suggesting that correct sentences elicited smaller P600 amplitudes. Furthermore, a significant interaction between semantic correctness and hemisphere were detected, $F(2, 52) = 5.19$, $p = 0.009$, $\eta_p^2 = 0.17$. This interaction indicated that correct sentences elicited smaller P600 amplitudes in the left ($\Delta = -1.24$, 95% CI [$-1.86$–$-0.61$], $p < 0.001$) middle ($\Delta = -0.69$, 95% CI [$-1.28$–$-0.11$], $p = 0.02$), and right areas ($\Delta = -0.44$, 95% CI [$-0.86$–$-0.02$], $p = 0.04$).

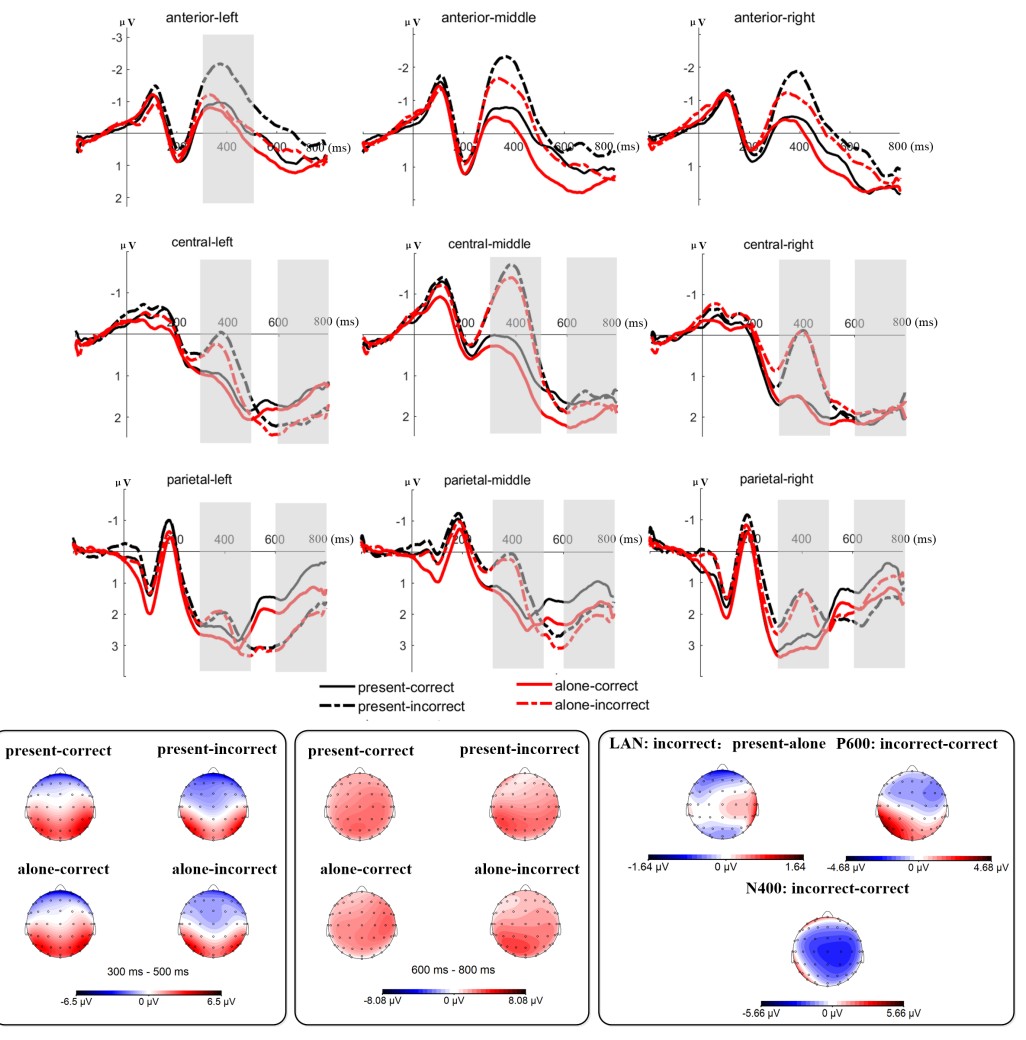

**Figure 3  Grand average waveform of ERPs in different sites.**

## DISCUSSION

This study investigates the impact of others' presence on Chinese sentence comprehension and its neural mechanism. The research manipulates semantic correctness, distinguishing between semantically correct and incorrect sentences. Participants read both sentence types alone and in the presence of others, during which electrical brain responses to key words were recorded. After reading, participants assessed the sentences' semantic accuracy. While in the company of others, participants were informed that semantically correct sentences had a higher accuracy rate than incorrect ones; however, the presence of others did not affect these rates. EEG results revealed that processing semantically incorrect sentences in the presence of others elicited a larger LAN in left frontal lobe compared to the alone condition. In addition, semantically incorrect sentences generated greater N400 amplitudes in the mid-posterior brain region and a larger P600 component in the rear of the brain.

At the behavioral level, this study found no effect of the social situation—"the presence of others"—on text reading, aligning with the findings of *Hinchcliffe et al. (2020)*. They positioned key words mid-sentence, requiring readers to process the entire sentence before making a judgment. The researchers proposed that this setup might allow for counteracting or compensating mechanisms, hence no detectable influence of others' presence on text reading. To mitigate this potential effect, this study positioned key words at sentence ends, allowing complete sentence processing post-reading and prior to judgment. The interval between the appearance of key words and judgment was insufficient to affect the influence of others' presence on text reading. No interaction between social context and semantic judgment was observed at the behavioral level, possibly due to the simplicity of the task, resulting in a ceiling effect. In processing semantically incorrect sentences, the presence of others induced a larger LAN component in the left front of the scalp than in their absence—a novel finding of this study. Although a LAN component was also observed in *Hinchcliffe et al.'s (2020)* experiment, significant differences exist.

First, the LAN components identified in the two studies occurred in different locations. This study observed an enlarged LAN in the left prefrontal area when participants read incorrect sentences in the presence of others. Second, the tasks triggering LAN components differed. Here, participants judged the semantic correctness of the sentences they read. Third, the experimental materials varied. This study employed a consistent syntactic structure, simple and strictly controlled for syntactic accuracy, devoid of syntax errors or morphological violations. Consequently, the LAN observed here is not induced by morphological or syntactic violations or complex syntactic structures. The increase in LAN amplitude can solely be attributed to the influence of others' presence. Studies on syntax have found that morphological/syntactic violation does not induce LAN waves, but induces N400 with larger amplitude (*Molinaro, Vespignani & Job, 2008*; *Mancini et al., 2011*), the possible reason is the transformation of processing strategy from algorithmic strategy to heuristic strategy (*Jiménez-Ortega et al., 2017*). We hypothesize that when participants encounter semantically incorrect sentences in the presence of others, due to clear semantic violations, they adopt a more conservative strategy to extract detailed information from the sentences. Based on these results, we speculate that in the presence of others, semantic integration becomes challenging, and individuals actively utilize both semantic and syntactic information for sentence reanalysis, with syntactic processing aiding semantic integration. In conclusion, LAN typically reflects morphological/syntactic processing difficulties, and we lack sufficient evidence to assert that it reflects semantic processing. This phenomenon warrants further discussion.

This study also found that, compared with semantically correct sentences, semantically incorrect sentences induced greater N400 amplitude in the middle and posterior brain regions, consistent with previous studies (*Hinchcliffe et al., 2020*). The increased N400 amplitude suggests that the semantic integration of incorrect sentences are more challenging. No interaction between social context and semantic correctness was observed. Participants were informed that their partners were to complete a memory task, while they themselves assessed the semantic accuracy of sentences. Thus, the tasks performed by the participants and their partners were independent and did not influence each other.

In contrast, in studies demonstrating the social N400 effect, participants engaged in joint tasks with their partners, requiring them to process speech from another's perspective (*Jouravlev et al., 2019*). Although participants could access background information and correctly interpret semantics, their peers lacked this information. Therefore, in scenarios where others were present but not engaged, no social N400 effect was evident. The P600 differences observed were confined to the processing of semantically incorrect sentences, indicating that semantic analysis and reintegration remained unaffected by the social context. This implies that the mere presence of others does not necessarily exacerbate conflicts between current input and the communicator model.

There are some limitations to this study. First, it focuses solely on the impact of others' presence on semantic processing and not on syntactic processing. Syntactic processing is an important part of sentence processing, and LAN is an important indicator of morphological/syntactic errors. This study found that semantic processing in the presence of others induced a larger amplitude of LAN, which cannot separate semantic and syntactic effects. Secondly, the positioning of the partner was not regulated; they were only permitted to sit to the right of the participant. The location of the partner is not matched left and right, and location interference cannot be removed from the ERP results. Subsequent studies should examine different peer positions to assess their effect on text reading. Third, this study did not explore participants' perceptions of their peers' roles within the experiment. Individual differences and different cognitive styles of the subjects may have additional effects on the experimental results. Finally, this study did not manipulate the order in which others appeared, but only balanced it as a control variable. Completing tasks with others first may cause individuals to devote more attention resources, affecting the outcome of subsequent tasks completed alone. Future research could explore how varying the order of others' presence impacts text reading.

## CONCLUSIONS

The presence of others elicits a stronger neural response during the initial stages of Chinese sentence reading. However, it does not impact semantic integration or conflict resolution. These results support the notion that contextual sociality influences text reading.

## ACKNOWLEDGEMENTS

We are deeply indebted to PhD. Hanwen Shi for helpful suggestions and comments on ERP analyses, and we thank Professor Yahong Li for valuable comments on previous versions of the manuscript.

### Funding

The following grant information was disclosed by the authors: National Education Science Plan (BBA230066). The funders had no role in study design, data collection and analysis, decision to publish, or preparation of the manuscript.

## Grant Disclosures

The following grant information was disclosed by the authors:
National Education Science Plan: BBA230066.

## Competing Interests

The authors declare there are no competing interests.

## Author Contributions

- Teng Yu conceived and designed the experiments, performed the experiments, analyzed the data, prepared figures and/or tables, authored or reviewed drafts of the article, and approved the final draft.
- Xue Sui conceived and designed the experiments, authored or reviewed drafts of the article, and approved the final draft.
- Yu tong Li conceived and designed the experiments, analyzed the data, prepared figures and/or tables, and approved the final draft.

## Human Ethics

The following information was supplied relating to ethical approvals (i.e., approving body and any reference numbers):

The Institutional Review Board of Liaoning Normal University approved this research (LL2024114).

## Data Availability

The data is available at OSF and Zenodo:

- Sui, Xue. 2024. ''Others Presence.'' OSF. November 22. osf.io/xvf8r.

- Yu, Teng. (2024). Presence of others dataset [Data set]. Zenodo. https://doi.org/10.5281/zenodo.13954526

## Supplemental Information

Supplemental information for this article can be found online at http://dx.doi.org/10.7717/peerj.18798#supplemental-information.

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
