# Peer review of "Social presence effect in language comprehension: evidence from event-related potential (ERP) research"

_PeerJ, doi:10.7717/peerj.18798_

## Round 0.1 · original submission · Major Revisions

We have received three thorough reviews of your paper. I perceive that all reviewers see merit in your research. It looks to me that the reviewers have provided feedback that should help you to significantly improve your manuscript.

When doing your revision, please also produce a separate 'response' document that provides a point-by-point response to reviewer feedback. Please indicate specifics around what has been changed, and also please include rationale for any points where authors have not made changes based on the feedback.

Based on my read of the reviewer feedback I recommend paying particular attention to the following:
- All reviewers have flagged that clarity of writing could be better. I encourage the authors to endevour to improve clarity and flow of wiritng for the resubmission wherever they can in the manuscript.
- Both R1 and R2 have flagged that the introduction of the role of presence could be fleshed out a little more. I note that R2 has provided some specific suggestions for improvement.
- R1 has flagged that your research is very similar to Hinchcliffe et al. (2020) and reocmmended that study be more integrated into your introduction and discussion sections which seems like very useful feedback to my mind.
- R3 has flagged that the rationale for hypotheses is thin, and I note that R1 has also flagged this and provided some specific suggestions for improvement.
- R3 has mentioned in their opinion that the introduction needs 'a lot more detail'. I can see how some additional details in the introduction would be useful, but I also encourage the authors to be as succinct as possible to avoid the introduction becoming too lengthy. I think taking on R1's suggestion to position the paper as following on from Hinchcliffe et al. (2020) should help to provide a clearer rationale for the work to avoid the need to make the introduction section too lengthy.
- R3 has mentioned that the Rueschemyer (2015) paragraph in the introduction was a bit difficult to read. I agree with their sentiments and I see they make some good suggestions for improvement.
- R3 has flagged that
- All reviewers have flagged that in the discussion of findings some explanations are perhaps not as compelling as they could be. I note reviewers have provided some suggestions to help for you to consider. I recommend making sure you are very clear when making the revisions what explanations are backed up well by pre-existing evidence and what explanations are more speculative and require future research to confirm or deny.
- R2 and R3 have made some comments about conducting further analyses. I see such things as optional for the authors to consider based on whether they feel doing such things would enhance the paper, or instead just add some unnessary extra complexity. For the issues raised that the authors don't carry out the extra analyses, such things could instead be raised as a limitation and future research talking point in the discussion.
- R2 has flagged that some things in the methods section could be made clearer via further clarifications, with specific questions provided.
- R1 and R2 have flagged that for the uploaded data, it would be nice to also include an excel file alongside your SPSS file. This is easy to obtain via SPSS using 'File' -> 'Export' (then choose an option). I think CSV format or tab delimited format options are probably best as those file formats are even more widely accessible than a standard excel file.
- A comment from myself... the authors may wish to consider clarifying some of the jargon terms (e.g., N400, LAN, etc) within the introduction by writing with the intention that your paper will be viewed by a more general audience, as PeerJ is a more 'generalist' kind of journal with a broader readership compared to more specific topic journals. I consider this an optional suggestion for you to consider, as based on the reviewer comments I anticipate that it will already be a challenge to accommodate the reviewer suggestions while also maintaining a relatively concise introduction section.

Reviewer 1 ·

Basic reporting

There are a few issues with regard to basic reporting.

1 -- The writing is generally good, but there are places throughout the manuscript where words are used inappropriately (e.g., line 137, "...divided averagely to two lists." --> maybe say "...divided across two lists."). Also, note that line 287 has a sentence fragment ("While recording...responses.").

2 -- The literature review on social presence effects needs to be improved. The review provided (lines 40-50) is rather cursory, and doesn't capture the complexity of the findings. In addition, it is never quite clear what the aspects of the literature that are mentioned have to do with the rationale for the current study (beyond saying that social presence effects exist).

3 -- The study seems to be a replication of Hinchcliffe et al. (2020), but this does not come through clearly in the introduction. The authors could spend more time discussing the details of Hinchcliffe et al.'s study and results, and what they expect from their replication.

Experimental design

I have a few comments on the experimental design.

1 -- The research question is not well-specified in the manuscript. The rationale for the current study is discussed at the end of the introduction (lines 100-107). It appears that the rationale for the study is that the authors want to know if social presence effects arise in Chinese. Replicating effects in different participant samples (and, for psycholinguists, replicating in different languages) is a worthy goal, but I wonder if the authors could say more about their purpose for conducting the study. Do they expect that the effects will be different in Chinese? Can they be more specific about the results they expect to find?

2 -- Related to the first point and comment #3 under Basic Reporting, it appears that this study is a more-or-less direct replication of Hinchcliffe et al. (2020). If that is the case, I think it should be highlighted at the end of the introduction. The authors should describe Hinchcliffe's study and results in more detail, and should also outline their expectations about what they expect to find. Is there any reason to expect that they will find something different than Hinchcliffe did?

3 -- I had no difficulty accessing the data and reading the files into SPSS. I also replicated the results of some of the analyses reported in the manuscript. Nevertheless, the documentation of the files needs to be improved. The authors should provide a key that defines each column in each of the data files to make it easier to navigate the files and reproduce the results. In addition, it would be good if the data were provided in Excel worksheets (or, a similar format) rather than SPSS files, as the Excel sheets will be more accessible to more users (since some researchers may not have access to SPSS).

4 -- On line 197-198, we are told that RTs were excluded when they were more than 2.5 SDs from the mean. Please specify whether this was the overall mean RT for that participant, the mean RT within a specific condition in the experiment, or something else.

Validity of the findings

1 -- The authors' discussion of their results seems to go beyond what could be concluded from the data that are presented. For example, lines 301-306 present the idea that participants worried about their partner's evaluation of them. Maybe, but what is the evidence for this? It seems like this is something that could be measured in the study (e.g., asking participants about this after the main tasks are complete). Similarly, the reasoning behind the lack of a social N400 effect is fine as far as it goes, but it seems that having a sense of the participants' perception of the experiment is crucial -- do we know how the participants perceived their partners and their role in the study?

2 -- As before, I think presenting the study as a replication of Hinchcliffe et al. would be helpful for framing the purpose of the study and the interpretation of the results.

Additional comments

As noted above, I think the paper could be re-framed as a replication of Hinchcliffe et al. (2020) and the results could be interpreted in that context. The authors might also consider conducting a follow-up study in which they probe the participants' understanding of the task and interpretation of their partner's presence to get a better understanding of what is driving the effects (or lack thereof) that are observed.

Reviewer 2 ·

Basic reporting

In this study, Sui and colleagues examined differences in ERP amplitudes – specifically the LAN, N4, and P6 components – when reading sentences either alone or in the presence of another in order to test the impact of social presence on language processing. Semantically anomalous sentence endings produced the usual effects – a larger N400 and P6 response – but the LAN in particular was larger following semantic anomalies when in the presence of another.

The study is well-motivated and the question is interesting, and the methods here are fairly sound - though I have issues which I believe must be addressed, which I have outlined below. I found the overall writing to be a bit unclear, perhaps due to English language issues, as well as an over-abundance of jargon. For instance, the 3rd paragraph of the Introduction section is rather dense and difficult to follow, with lots of study specifics that would be better summarized. Additionally, there are various grammatical errors (e.g., on page 8, “the confederate would memory the presented word”, should be “memorize the presented word”). To improve the quality of the manuscript and communicate the important ideas better, I would personally suggest that the authors consider having a colleague who is proficient in English and familiar with the subject matter review the manuscript, or contact a professional editing service.

I appreciate that the authors have shared the data - however, the data is only shared in SPSS format. If possible, it would be better to share the data in a more accessible format.

Experimental design

There are a few areas where the design and the methods fall short. Namely:

1) More information is needed regarding the stimuli used in the experiment. How was “semantic correctness” determined? What were the plausibility and predictability of the critical words of the sentences? The authors state that the number of strokes and frequency of the critical words were matched across different lists – how was this determined? If this was through a statistical test, this should be reported.

2) More description is needed for the ICA correction of ocular artifacts, as currently only a sentence is provided. What algorithm was used? How were components selected? How many were removed across participants? Were the topographies indicative of blinks as well as saccades?

Validity of the findings

There are some issues with the findings, namely with the LAN effect. I outline these below:

1) While the left lateralization of the LAN effect is consistent with previous literature, the topography of the LAN effect presented in the results is concerning and potentially more indicative of eye movements. This is additionally concerning since, in the social presence condition, the confederate always sat to the right of the participant, and the participant could be making a glance in this direction. The experimenters could have potentially had the confederate sit on the left or right, and counter-balance the placement of seating, but it doesn’t appear this was done. Furthermore, it appears that only vertical eye movements were monitored, and not horizontal eye movements, and participants were only told not to blink. The presence of another person in peripheral vision could encourage saccades in this direction. Without horizontal eye channels, it is difficult to know if saccades have contributed – however, the authors might be able to visualize this with a bipolar channel of the most frontal left and right channels (e.g., either FP1-FP2, or AF7-AF8). This should be plotted across conditions to determine if there were more saccades in the error condition with the confederate present.

2) Following the point above, more discussion is needed regarding the LAN effect reported here. To my knowledge, LAN responses have been largely attributed to morpho-syntactic processing, and are elicited by agreement violations (e.g., “The two bugs hum / hums loudly when flying”), which can be dissociated from the semantic processing of the N400. Firstly, the sentences should be checked to determine if agreement violations. Second, at a theoretical level, if the LAN reflects morpho-syntactic processing, why would the LAN response be elicited here, and be enhanced in the presence of a confederate? The statement of “We speculate that the presence of others may change the individual's language processing and enhance the syntactic processing” is not enough of a discussion of this result.

As an additional point, the authors counter-balanced the order of the social conditions (alone vs. present) across participants; if possible, I believe this factor should be analyzed, perhaps by comparing the ERP waveforms for alone first subjects to alone second subjects. It seems to me that having some practice doing the task first alone could impact the responses that occur when doing the task with someone else present.

Reviewer 3 ·

Basic reporting

1. The manuscript could benefit from much clearer, less ambiguous language throughout, as well as more professional English phrasing so that an international audience can clearly understand the research. There were many instances in the paper where the wording was unclear or confusing, rendering it difficult to understand the author’s point(s). I will list some of these instances below:

- “The sociality of speech use” (line 34) – it is unclear what the authors mean here
- “and are tested to be difficult to process” (line 83-84) – again, I am not sure what the authors mean here
- “However, during syntactic processing, other presence induced greater amplitude LAN in the mid-posterior brain region than that induced by presence alone.”(lines 92-94). One thing that would help the reader throughout the manuscript is rephrasing the term “presence alone”. I believe the authors are referring to various conditions in experiments where participants are alone, i.e. not in the presence of others. The term “presence alone” is quite confusing; I recommend referring to these conditions as just “alone” since the word “presence” implies that someone else is present.
- “If not, the sentence referred to the semantically incorrect sentences. if the critical words did not matched the sentence meaning, the sentence was defined to the semantically incorrect sentences” (lines 133 – 135). I can’t decipher what the authors mean here.
- “These experimental sentences were divided averagely to two lists.” (lines 136-137). It is unclear what the authors mean by “divided averagely”
- “Therefore, each list contained 50 semantically incorrect sentences, 50 semantically incorrect sentences and 25 filler sentences.” (lines 138-139). I believe one of the “semantically incorrect” should say “semantically correct”
- The authors refer to their semantic violation condition using a number of different phrases (semantically unreasonable, semantically irrational, IMPLAUS). I would recommend being consistent and more clear by using the same phrase throughout so that the reader can follow.
- “There two ANOVAs performed on central and posterior sites severally” (lines 213 to 214). It is unclear what this sentence means, which makes it difficult to follow the statistical procedure.

The manuscript contains the following misspellings:

- Line 75, Rueschemeyer
- Line 90, synthetically (should be syntactically?)
- Line 121, add a space after “University”

I recommend that the authors have their manuscript reviewed and edited by someone (either a colleague or even a professional editing service) who is proficient in English. I think editing the manuscript’s language would go a long way for this paper.

2. The introduction needs a lot more detail. The authors discuss a few theories of how the presence of others affects us affectively as well as cognitively. The authors also scantly survey how the presence of others affects language processing, but this section is missing some key areas of literature. For example, the authors do not discuss the role of perspective-taking in language comprehension (for a review, see Barr & Keysar (2006)), mutual knowledge (Clark & Marshall (1981), or the debate between ego-centricity (Keysar (2007)) and perspective-taking literature. Given that language is primarily processed and produced in the presence of others, and that it has evolved and is learned through social interaction, the presence of others is fundamental to our language processing. The authors do not sufficiently address this point.

The full citations for the papers I have referenced above listed below:

Barr, D., & Keysar, B. (2006). Perspective taking and the coordination of meaning in language use. In M. Traxler & M. Gernsbacher (Eds.), Handbook of psycholinguistics (pp. 901–938). Amsterdam: Elsevier.’

Clark, H. H., & Marshall, C. R. (1981). Definite reference and mutual knowledge. In Elements of discourse understanding (pp. 10–63).

Keysar, B. (2007). Communication and miscommunication: The role of egocentric processes. Intercultural Pragmatics, 4(1), 71–84. https://doi.org/10.1515/IP.2007.004

Another relevant paper:

Brennan, S., Galati, A., & Kuhlen, A. (2010). Two minds, one dialog: Coordinating speaking and understanding. Psychology of Learning and Motivation, 53, 301–344.

3. The authors should spell out their hypotheses at the end of the introduction. At the moment, without clear hypotheses and a discussion of the implications of their hypotheses, it is not clear why the current study is important, or what gap in the literature it fills.

4. I commend the authors for discussing the Rueschemeyer 2015 paper in detail, as it is very relevant to their study. That said, I find their review to be quite difficult to follow, particularly the third paragraph of the introduction. I would recommend keeping the summary of the paper to a high-level of detail, taking out the condition names (PLAUS, IMPLAUS, CONTEXT), and sticking to the main points of the study. E.g. what was the research question, why does it matter for the current study, what did the study find, and what are the implications of these findings, particularly as they pertain to the current study. The authors conclude with “These results indicate that the presence of others affects speech processing, which proves the existence of social presence effect.” How, exactly, does the presence of others affect speech processing, and what does this say about theories of language? In general, the authors do not sufficiently introduce theories of language comprehension, and how processing language alone versus in the presence of others may differ.

5. The end of the introduction needs more details about what the literature cited means, i.e. what does it tell us about theories of language processing, why is it important, and why is it relevant? The authors do a decent job of bringing different literatures into the last paragraph by describing certain studies, but they do not assess what these studies tell us about language processing in the presence of other people, and why these studies are important to the current study.

6. The authors should introduce ERP components. At the moment, the manuscript mentions the N400 (line 77) without providing the reader with the context of what an ERP component is, and what the N400 is thought to index. Why is it interesting and relevant to the current study that the presence of another person who does not have access to the same linguistic context produces a larger N400 effect? What does this tell us about language processing? The social N400 studies that the authors cite show the very interesting phenomenon that the presence of another person, and your knowledge of what that person knows, affects how you comprehend words. This tells that we consider other people's perspectives within a few hundred milliseconds of encountering a word. I recommend that the authors focus on this overarching implication of the social N400 studies. I recommend that the authors also cite more papers, especially more recently published papers, when introducing the LAN and P600 components. Our understanding of the P600 has evolved since the mid 00s (e.g. Kuperberg et al., 2020), and this work should be reflected in the manuscript.

Full citation: Kuperberg, G. R., Brothers, T., & Wlotko, E. W. (2020). A tale of two positivities and the N400: Distinct neural signatures are evoked by confirmed and violated predictions at different levels of representation. Journal of cognitive neuroscience, 32(1), 12-35.

7. The caption for Table 1 states that “critical words have been bolded” but they are not bolded.

8. In general, the figures need more detail in their descriptions to explain what each component of the figure depicts.

Experimental design

1. The authors do not define their research question, do not state their hypotheses, and do not write anything about why the present research fills an identified knowledge gap. This is critical to any manuscript, and I strongly suggest that the authors include a paragraph at the end of their introduction speaking to these critical points.

2. I strongly recommend that the authors cloze norm and plausibility norm their stimuli. From the current manuscript, I gather than the authors used their intuition to determine whether stimuli were semantically coherent or semantically incoherent. It is critical to validate these intuitions by asking other participants to rate how semantically coherent the stimuli are using plausibility norming. Furthermore, the authors should cloze norm their stimuli. We know that the N400 component is inversely correlated with cloze probability, for references see below:

M. Kutas and K. D. Federmeier, "Thirty years and counting: Finding meaning in the N400 component of the event-related brain potential (ERP)", Annu. Rev. Psychol., vol. 62, no. 1, pp. 621-647, 2011.

M. Kutas and S. A. Hillyard, "Brain potentials during reading reflect word expectancy and semantic association", Nature, vol. 307, no. 5947, pp. 161-163, 1984.

G. R. Kuperberg, T. Brothers and E. W. Wlotko, "A tale of two positivities and the N400: Distinct neural signatures are evoked by confirmed and violated predictions at different levels of representation", J. Cogn. Neurosci., vol. 32, no. 1, pp. 12-35, 2020.

Without offline ratings of the stimuli, the validity of the present study cannot be established.

3. The authors should clarify whether they average their data over time windows and regions of interest. I gather from the manuscript that they do so, but this could be made more explicitly clear in lines 203-205.

4. The authors chose to conduct a repeated measures ANOVA using averaged ERP data over time windows and regions of interest (I believe…see comment above). This may not be the most suitable analysis for the current data set. This type of analysis does not always control for Type I error, and is inflexible (Fields & Kuperberg, 2018). As Fields and Kuperberg show, permutation‐based mass univariate tests can be employed with this type of design for greater power and flexibility. The authors should justify why they chose to conduct their analyses using the methods they do.

Validity of the findings

1. The authors do not discuss the rationale of their study, and implications of their findings, or the benefit to the literature. This should be clearly stated in the manuscript for the reader to understand the importance of any study.

2. The authors also do not clearly state their conclusions and do not link the conclusion to the original research question. Their conclusions are too vague. For example, they mention that “the presence of others may change the individual’s language processing and enhance the syntactic processing” (line 314). This statement is quite vague. What do the authors mean by “enhanced syntactic processing”, and how, exactly, does the presence of others change language processing? Similarly, the authors mention, in the conclusion, “the enhancement of semantic integration and reanalysis.” Again, this phrase is vague. What is meant by enhanced semantic integration and analysis? How does this inform current theory of language processing?

3. The authors refer to their results as pertaining to “speech process” several times in the Discussion section. For example, “Future research is necessary to explore the influence of other people’s presence on speech processing when subjects and others perform the same task” (lines 335 to 336). However, the current study is based on reading sentences. The link between sentence comprehension speech processing is not made, so it is not clear how the study’s results relate to speech processing.

4. I find that the manuscript is lacking in its discussion of limitations, as well as future directions. I recommend that the authors discuss the limitations in greater detail, specifically discussing how they could have affected the results, and include some future directions.

Additional comments

n/a

---

## Round 0.2 · Minor Revisions

Based on the re-review, prior to acceptance for publication, may I ask you to address the final minor points:

- Please look over R1 additional feedback and make any changes that you feel will further improve the manuscript.
- R2 has made an additional comment about changing some language in the discussion which looks like a fair point, and looks like a very minor edit for you to implement.

If you are able to handle the above minor points, then I don't think we will need to send out to review again.

Reviewer 1 ·

Basic reporting

The authors have made extensive revisions to their manuscript in response to the first round of comments. These revisions have improved the manuscript, but I think that there are still a number of issues that need to be addressed.

1 -- The new content in the introduction does not seem to fit with the overall purpose of the paper. For example, at the end of the first paragraph the authors say that the interplay between interlocutors is a key area of research that is often ignored. However, their research also ignores the interplay between interlocutors as the confederate and the participant are not engaged in conversation. As another example, the authors present a long list of differences between English and Chinese. This is fair enough, but they do not connect these differences to the current study. Why would any of these differences lead one to expect a different social presence effect?

2 -- I am not sure that some of the claims in the paragraph between lines 88 and 102 are justified. For example, the authors suggest that Indo-European languages have a separation between syntax and semantics, and that syntactic processing is the foundation of sentence processing. Is this the best way to characterize things, especially considering the way that the constraint-satisfaction approach to comprehension and the "good enough" approach to comprehension describe the comprehension process?

3 -- The authors refer to "speech processing" throughout the paper, but it struck me that in at least some cases they are using the term to refer to cases where participants are reading. For example, the authors use the term "speech processing" on line 293 when referring to their own study (which involves reading). Check through the manuscript to make sure that the terminology is used appropriately throughout.

Experimental design

1 -- I don't have specific comments here, but I will note that it is not clear that the authors really addressed Reviewer 3's point #4 in this section (about the use of ANOVA in their data analysis). I will leave it to the Reviewer to comment more fully on whether their response is satisfactory.

Validity of the findings

I have a few comments about the findings.

1 -- I followed the OSF link provided in the paper, and as far as I can tell the authors did not upload the excel files. I can still access the zip folder with all the SPSS files, but I could not find any other files on the site (and, it appears that no files have been added since July 9, 2024). Perhaps the authors can check to ensure that the link is correct, and make sure to upload the excel files.

2 -- I think that the authors don't really address Reviewer 2's concerns about the results. For example, they note that future studies could explore the effect of task order. Why not just analyze the current data set to see if task order matters? Furthermore, I'm not convinced that the text added by the authors addresses the Reviewer's concern about the interpretation of the LAN. I will leave it to Reviewer 2 to fully evaluate this point as I am not an expert on this area of the literature, but I didn't find the author's response to be compelling.

3 -- Starting at line 340, the authors list a long set of limitations to their study. I think it is fine to acknowledge limitations, but perhaps the authors can say something about why these issues might matter, and how they might affect the results.

Additional comments

After going through two rounds of review with this manuscript, it seems to me that the main issue is an overall lack of clarity about what the study is designed to accomplish. That may be why the new elements of the introduction do not help to ground the current study more firmly in the literature. Given Reviewer 2's concerns about the interpretation of some of the results, I wonder whether the authors wouldn't be better off conducting another experiment or two to address the concerns raised in the reviews and get a better picture of what is going on in their task.

Reviewer 2 ·

Basic reporting

The reporting in the manuscript has been improved, and the paper reads fairly clearly now.

Experimental design

The additional information added to the Methods clarifies the issues that I had. I thank the authors for clearing this up.

Validity of the findings

I don't think the potential issue of eye movements contributing to the results has really been clarified for me - it seems, from the authors explanation, that the confederates always sat on the right, and despite the text on the screen not requiring saccades to be read, the criticism was not regarding the stimuli on the screen, but rather participants glancing at the confederate, who was in their peripheral vision. The topographical plots aren't sufficient to address this issue - these were already present in the first version. The request I made of creating a bipolar channel to plot and analyze was specifically because it would better be able to show saccade-related activity.

However, the editor views such an analysis as optional, and I concede to the editor's wishes. The additional explanation for future studies in the Discussion is helpful.

Additional comments

I have a minor suggestion, which is perhaps a nitpick - the authors make neuroanatomical statements several times, particularly in the Discussion. For example, "a larger LAN component in the left frontal lobe". EEG does not have the spatial resolution to determine this - even if the component is over the left front of the scalp, we cannot say for sure it's generated by the left frontal lobe. I would edit these statements to be less specific to brain areas, and more to areas on the scalp.

---

## Round 0.3 · accepted · Accept

Thank you for making those further minor edits, and your response document that detailed what was done. Congratulations on your work being completed.

You can see that R1 has flagged some minor grammatical issues for you to consider when getting the paper ready during the final proofing stage that I recommend you consider.

Reviewer 1 ·

Basic reporting

I think the authors have done a good job revising the paper. I have just a few minor comments.

On Lines 46-7, the first sentence talks about text comprehnsion, then the next talks about speech comprehension. Check to make sure the labels are accurate.

Line 91 -- The sentence starts with "Second, ...", but there was no "First, ..." presented earlier. Maybe replace "Second..." with "In addition,..."

Line 314 -- This is an odd way to start a new paragraph. Maybe take the last line of the preceding paragraph and make it the first line of this new paragraph.

Experimental design

No remaining comments.

Validity of the findings

No remaining comments.

Additional comments

The authors did a nice job responding to the reviews. I recommended some minor changes, but overall I think the paper is acceptable for publication.